# Efficacy of Neuro-Feedback Training for PTSD Symptoms: A Systematic Review and Meta-Analysis

**DOI:** 10.3390/ijerph192013096

**Published:** 2022-10-12

**Authors:** Jian Hong, Jin-Hyuck Park

**Affiliations:** 1Department of ICT Convergence, The Graduate School, Soonchunhyang University, Asan 31538, Korea; 2Department of Occupational Therapy, Soonchunhyang University, Asan 31538, Korea

**Keywords:** post-traumatic stress disorder, PTSD, PTSD symptoms, neuro-feedback training, NFT

## Abstract

If the negative emotions experienced in life become trauma, they affect daily life. Neuro-feedback technology has recently been introduced as a treatment, but many different neuro-feedback protocols and methods exits. This study conducted a meta-analysis of neuro-feedback training for post-traumatic stress disorder (PTSD) symptoms to evaluate the effects of functional magnetic resonance imaging (fMRI) and electroencephalogram (EEG)-based neuro-feedback training. A search of PubMed, the Cochrane Library, Web of Science, Science Direct, and ClinicalTrials.gov was conducted from January 2011 to December 2021. The studies’ quality was assessed using the Cochrane risk of bias tool and publication bias was assessed by Egger’s regression test. Seven studies that met the inclusion criteria were used for the systematic review and meta-analysis. EEG was more effective than fMRI for PTSD symptoms, and the effect on PTSD symptoms was higher than on anxiety and depression. There was no difference in the effectiveness of the training sessions. Our findings showed that EEG-based neuro-feedback training was more helpful for training PTSD symptoms. Additionally, the methods were also shown to be valid for evaluating clinical PTSD diagnoses. Further research is needed to establish a gold standard protocol for the EEG-based neuro-feedback training (EEG-NFT) method for PTSD symptoms.

## 1. Introduction

Many individuals fail to endorse their negative reactions at the time of an event [1]. It is estimated that one in ten people who experience trauma develop symptoms of post-traumatic stress disorder (PTSD) [2]. Like PTSD, mood indicators can comprise a loss of interest or pleasure in activities, guilt, or loneliness [3] and have abnormal brain activity patterns [4]. PTSD is characterized by re-experiencing a traumatic event and negative alterations in mood and cognition [1]. PTSD symptoms affect personal and social life conditions [5], and PTSD symptoms may vary in severity depending on the level of the symptoms the individual feels. The association between PTSD and deficits in emotion regulation is well established [6]. PTSD is treated using four main therapeutic techniques [7], which include cognitive behavioral therapy, exposure therapy, eye movement desensitization and reprocessing [EMDR], and pharmacotherapy [8]. However, there is no single standard treatment for PTSD [7] and, especially in the case of pharmacotherapy, 41% of the patients who receive the treatment fail to respond [4].

For PTSD, self-regulation, increased neuroplasticity, and the recapture of the brain’s functional network are the primary factors which are important in healing [9]. Neuro-feedback (NF) is a relatively new technique that allows us to target certain brain areas for developing neuroscience-guided therapy [10]. It is a form of behavioral training aimed at modifying the skills needed for the self-regulation of brain activity [11]. Thus, in an alternative way, the use of NF avoids the potential triggering of negative experiences pertaining to the trauma [3]. Several studies showed that NF could be helpful for patients with PTSD [9]. By allowing us to self-regulate our brain activity [12], neuronal activity and connectivity can be changed [8], indirectly resulting in behavioral changes [5]. Brain activity can be fed back in multiple ways using several neurophysiological methods (e.g., real-time functional magnetic resonance imaging [fMRI], magnetoencephalography [MEG], functional near-infrared spectroscopy [fNIRS], positron emission tomography [PET], and electroencephalogram [EEG]) [13,14]. These feedback techniques allow the participants to monitor, interact with, and manage their mental states [15] and current states of cognition [16] by rewarding desirable patterns of brain activity with visual, auditory [17], or representational forms such as graphs, numbers, video games, and moving objects [18].

PTSD symptoms derive from several areas of the brain. Abnormal contextual processing is often observed in patients with PTSD and it relies on the functional integrity of the medial prefrontal cortex (PFC), hippocampus, and amygdala [19]. A previous meta-analysis found that the medial PFC, including the amygdala and hippocampus, was implicated in PTSD [20]. Maild and colleagues (2009) showed that patients with PTSD had more amygdala activity and less activity in the PFC and hippocampus than healthy individuals [9]. The PFC is important for the regulatory capacity [9] and an increased amygdala activation was seen in all trauma-exposed individuals [6]. Additionally, PTSD symptom reduction was associated with an increase in anterior cingulate cortex (ACC) activation [6]. Kohn et al. (2014) considered the dorsal ACC (dACC) as a core node for emotion regulation [6]. In resting-state studies of PTSD, enhanced slow waves in the left temporal region and decreases in the right parietal cortex were reported [21].

Many different NF protocols and methods exist [4]. NF by both EEG and fMRI represents an emerging adjunctive treatment that allows participants to self-regulate their current neural states [22]. fMRI-NFT is a novel method [23] used to identify the associated brain structures [4] and EEG-NFT is another way to obtain information on the functional neuroanatomy related to clinical symptoms [4]. fMRI-NFT increases or decreases activity in specific cortical areas and has been used to modulate the neural correlates underlying psychopathology [22]. fMRI-based NF training (fMRI-NFT) for PTSD has also been used to modulate amygdala activity [24] and the ACC. In contrast, EEG-NFT is used to regulate more global neuro-signals, indicative of larger-scale brain oscillations [22]. EEG-based NF training (EEG -NFT) for PTSD is mainly used to regulate the power of alpha waves alone or both alpha and theta waves [24]. According to an EEG meta-analysis study, NF treatment and protocol characteristics were different in previous studies: some delivered daily treatment and others less frequent sessions, the duration of treatment varied, and the number of sessions and type of NF differed [8]. However, a previous study [23] reported that the clinical effect of NFT is still missing. Additionally, these two methods differ in NF, but the difference in their effectiveness is still unclear. Therefore, the present study aimed to evaluate the effects of fMRI-NFT and EEG-NFT with PTSD participants compared to sham-NFT and no intervention. The primary outcomes were PTSD symptoms, anxiety, and depression.

## 2. Materials and Methods

### 2.1. Search Strategy

The studies were identified through comprehensive searches of PubMed, the Cochrane Library, Web of Science, Science Direct, and ClinicalTrials.gov databases conducted from January 2011 to December 2021. The search string was structured using the PICOS method: P (population) = the patients for PTSD, I (intervention) = neuro-feedback training, C (comparison) = sham, other cognitive training, control group, or no intervention, O (outcome) = PTSD symptoms, anxiety, depression, and S (study design) = randomized controlled trials (RCTs). The search terms were trauma OR (PTSD OR post-traumatic stress disorder) AND (neurofeedback training OR NF-training OR NFT).

### 2.2. Inclusion and Exclusion Criteria

We combined the search results from five different databases and deleted duplicate records. Studies were included if they met the following criteria:(1)Language: English;(2)Design: RCT;(3)Intervention: fMRI-NFT, EEG-NFT of alpha activity, e.g., alpha peak amplitude, alpha-theta ratio, or alpha and theta activities;(4)Control group: PTSD patients who received other treatments, sham-NFT, or no intervention;(5)Participants: PTSD patients;(6)Evaluation: assessments of PTSD, anxiety, and depression.

Studies were ineligible if they were not written in English or were conference abstracts or were not full texts.

### 2.3. Quality Evaluation

The studies’ quality was assessed using the Cochrane risk of bias (RoB) tool (Higgins and Green, 2011) in randomized controlled trials (RCTs). We determined the quality of each study according to the selection bias, allocation, detection bias, performance bias, attrition bias, and reporting bias. Three levels (low, unclear, and high) of a risk of bias were evaluated. To determine if there was any bias in the individual results included in this study, publication bias was assessed by the visual inspection of funnel plots and by Egger’s regression test (two-tailed *p*-values of <0.05 were considered to indicate the presence of the bias).

### 2.4. Data Extraction and Statistical Analysis

Data were extracted in duplicate using standardized data extraction forms. Information was gathered on the study’s design, participants, NFT characteristics (type, modality, and sessions), and results. The outcomes were PTSD diagnosis, anxiety, and depression in PTSD participants. Studies were excluded if they met the following criteria:(1)No control group design;(2)Intervention: cognitive tasks with no NFT, no EEG-NFT, or no fMRI-NFT;(3)Control group: healthy population;(4)Evaluation: data without assessment tools.

All analyses were conducted using Comprehensive Meta-Analysis 2.0 software [25]. We analyzed the total effect on PTSD symptoms (PTSD diagnosis, anxiety, and depression) using Hedges’ g with 95% confidence intervals (CIs). Hedges’ g is the index of an effect adjusted for any intervention differences between experimental and control groups (Hedges’ g < 0.3 indicates a small effect and g > 0.6 indicates a high degree of an effect size) [25]. The I-squared statistic was used to quantify heterogeneity in each included study, with values of 25, 50, and 75% reflecting small, medium, and high heterogeneity, respectively [17]. I-squared > 50% and *p* < 0.05 indicated notable heterogeneity, and a random effect model was used [25]. Second, EEG and fMRI studies were divided and their effect on PTSD was analyzed. Third, we used meta-regression to explore the effect of NFT sessions’ length on the overall PTSD symptoms.

## 3. Results

### 3.1. Search and Exclusion Results

Figure 1 shows the PRISMA flow chart for the study selection process. A total of 1174 records were identified from the initial database searches. After removing the duplicates, 40 records were screened in the first screening step. The eligibility criteria and a second screening step using PICOS identified 10 records. A total of seven records fulfilled the inclusion criteria and were finally included in this study.

### 3.2. Description of Included Studies

A total of seven records met the criteria for qualitative synthesis (Table 1). Studies with an RCT were published from 2016 to 2021. There were two from PubMed, seven from the Web of Science, and three from the Cochrane library, of which a total of 9 were duplicated. Three of the included studies used fMRI-NFT and four used EEG-NFT. Overall, all studies included PTSD participants. The intervention protocols, with differences in NFT type, modality, and NFT-sessions are shown in Table 1. The recorded brain regions and electrodes were the left amygdala (fMRI-NFT) and alpha activity (8–12 Hz) or the theta (4–7 Hz)-alpha (8–12 HZ) ratio (EEG-NFT). Of these studies, three studies used left-amygdala NFT, two studies used alpha NFT, and two studies used the theta-alpha ratio. The number of NFT-sessions ranged from 3 to 25 and the duration of the NFT-sessions was in the range of 6–40 min. PTSD symptoms including PTSD diagnoses were measured as the primary outcomes in all studies. The following tools were used: the Clinician-Administered PTSD Scale(CAPS), PTSD Checklist-Military Version(PCL-M), PTSD Diagnostic and Statistical Manual of Mental Disorders Checklist-5th edition(PCL-5), the Davidson Trauma Scale(DTS), the Montgemery–Asberg Depression Rating Scale(MADRS), Beck Depression Inventory(BDI), Hamilton Anxiety Scale(HAM-A), Hamilton Depression Rating Scale(HDRS), Beck Anxiety Inventory(BAI), Impact of the Event Scale-revised(IES-R), and Inventory of Altered Self-Capacities(IASC). This study was approved by the local research ethics committee and registered at the RROSPERO ID: CRD42022355878.

### 3.3. Synthesis of Results

#### 3.3.1. Overall PTSD Symptoms

Figure 2 and Figure 3 show the effect size of NFT using EEG and fMRI on PTSD symptoms in seven studies. The result showed a significant overall effect on PTSD symptoms (Hedges’ g = −0.789, 95% CI: −1.004 to −0.392, *p* < 0.05) with a high heterogeneity (Q = 18.286, *p* = 0.006, $I^{2}$ = 67.188). When divided into NFT types (EEG or fMRI), a significant result was obtained from EEG-NFT (four studies, Hedges’ g = −1.132, 95% CI: −2.061 to −0.203, *p* < 0.05). In contrast, fMRI-NFT (three studies, Hedges’ g = −0.368, 95% CI: −0.851 to 0.115, *p* < 0.05) showed low heterogeneity (Q = 0.156, *p* = 0.925, $I^{2}$ = 0.000), but the effect size was not significant for PTSD symptoms. Figure 4 shows the funnel plot of overall PTSD symptoms. One out of seven of the included studies was skewed from the mean to left. Publication bias was not found (*p* > 0.05). 

#### 3.3.2. PTSD Diagnosis

Of these PTSD diagnosis assessments, CAPS, PCL-5, and DTS were from EEG-NFT and CAPS and PCL-M were from fMRI-NFT. The effect and heterogeneity tests showed that there was no heterogeneity (Q = 2.930, *p* = 0.711, $I^{2}$ = 0.000), and the effectiveness of NFT on the PTSD diagnosis evaluation was significant (Hedges’ g = −0.658, 95% CI: −0.983 to −0.333, *p* < 0.05) (Figure 5). Figure 6 shows a result of publication bias for PTSD diagnosis, but there was no bias (*p* > 0.05).

#### 3.3.3. Anxiety and Depression

Anxiety assessments were made using MADRS and BDI. Depressions assessments were made by HAM-A, HDRS, and BAI. There was a high heterogeneity (Q = 12.587, *p* = 0.013, $I^{2}$ = 68.221). The lower the dependent variable, the greater the effect size. The effect size was not significant for anxiety and depression (Hedges’ g = −0.562, 95% CI: −1.230 to 0.106, *p* > 0.05) (Figure 7). However, there was no publication bias for anxiety and depression (*p* > 0.05) (Figure 8).

#### 3.3.4. Meta-Regression by Dose Effects

All of the studies included in this paper experimented with NFT for PTSD participants, and the length of the NFT sessions was different. Therefore, the sessions’ length was broadly divided into short sessions and long sessions. There were three to seven sessions in the short-session group and 16 to 25 sessions in the long-session group. Figure 9 shows a graph of the correlation analysis of the effectiveness of NFT according to the number of sessions. Although short-sessions tended to be more effective, the effect in the regression analysis was not significant (slope = −0.32581, *p* = 0.29661).

### 3.4. Risk of Bias in the Results

Figure 10 shows a summary of the RoB assessment. All seven studies were at low levels for data completeness and result reporting, and at a low risk of bias. Most of the allocation and blinding of personnel and assessments showed an unclear or low risk. However, two studies showed a high risk of bias, one study for the blinding of personnel (the experiment was conducted even though the participants recognized that they were in the experiment group) and one study for the blinding assessments (staff knew which group the participants belonged to).

## 4. Discussion

This study aimed to analyze whether NFT was effective in PTSD participants by dividing methods into EEG and fMRI as representative techniques, and whether there was a clinical improvement effect in a diagnosis evaluation and an anxiety and depression evaluation of PTSD symptoms. We found that EEG-NFT was more effective in improving PTSD symptoms than fMRI-NFT, and the effectiveness of NFT was significant in the scores seen in the PTSD diagnostic evaluations.

A previous study [23] reported that the clinical effect of NFT is still missing. There is no gold standard NFT protocol. The results of this study found that fMRI-NFT did not significantly affect clinical PTSD symptoms. Additionally, Nicholson et al. (2020) [22] reported that there were differences in the neurobiological mechanisms of EEG and fMRI real-time NFT. fMRI-NFT showed high emotional responses in the PFC as the activity of the amygdala increased, so the connection between the PFC and the amygdala was the main brain region target, whereas in the case of EEG-NFT, PTSD symptoms were reduced by connectivity with the amygdala through the regions that treat processing and trauma memory and emotion regulation in the PFC. It is said that the process of processing information about fear and trauma has an indirect interaction with the amygdala. Therefore, it seems to have a positive effect on the PFC interaction with the amygdala. In addition, according to the last systematic review for EEG-NFT [33], EEG helped to reduce stress levels by regulating alpha activity because it improved the working memory or the speed of visual information processing. Stress is a common symptom in patients with PTSD and EEG, which is effective in relieving stress, is also considered to be helpful in reducing PTSD symptoms. Additionally, in the recent EEG study for PTSD, the theta band of PTSD patients was significantly reduced [34].

PTSD causes cognitive and emotional problems, including negative self-beliefs and expectations and an inability to experience positive emotions [35]. There is a high level of comorbidity among PTSD participants, especially depression and anxiety [4]. To evaluate PTSD symptoms, the effects of trauma and adverse life events are usually investigated using clinical interviews and self-reported instruments [36]. In this study, a tool for diagnosing PTSD according to the cutline score calculated by measuring the level and sensitivity of the PTSD level, as well as anxiety and depression, were considered as psychological factors. As a result, PTSD diagnosis showed significant differences pre- and post-NFT, but not in psychological and physiological responses such as anxiety and depression. According to a study on PTSD [37], Roemer, Orsillo, Borkovec, and Litz (1998) found that PTSD severity was not necessarily significantly related to fear or other negative emotions caused by trauma. Thus, stress from PTSD causes fear, anxiety, and depression, and these side reactions are not all related to predicting PTSD. PTSD is associated with PTSD symptoms, such as the sensitivity accepted by individuals, where they suffered trauma, what type of trauma they experienced, and how they responded to their emotions. The same study reported that an individual’s emotional response should be measured in a different way than using self-reported measures to evaluate somebody’s emotions. Of course, responses according to negative emotions can be part of PTSD symptoms, but it is not responses that are uniquely visible to every PTSD.

Studies on NFT as a method of treating PTSD were analyzed. Twenty-four sessions of EEG-NFT led to significant improvements in PTSD symptoms and emotion regulation [35]. Nicholson et al. (2020) [37] evaluated patients over a long period and found that 61.1% of the participants in the NFT group no longer met the criteria for PTSD. This finding suggested that long-term PTSD treatment would be required. However, in this study’s analysis, there was no effect on PTSD symptoms following short and long NFT sessions

Taken together, in the current study, out finding showed that EEG-NFT was more likely to affect PTSD symptoms than fMRI, and NFT was found to be more effective in PTSD diagnosis than anxiety and depression. However, care should be taken with the interpretation of our results. Because of our small sample size, the results of the publication bias and the effect size are less reliable [38].

Since the gold standard protocol for NFT used to treat PTSD symptoms remains unclear, future studies are needed to examine the effectiveness of different sessions, brain target-regions, and training protocol methods on PTSD symptoms.

## 5. Conclusions

We explored the effect of neuro-feedback training on PTSD symptoms. Our findings indicated that EEG-based neuro-feedback training was more helpful to train PTSD symptoms than fMRI-based neuro-feedback training. Moreover, this effect suggests that it is also valid for the evaluation of PTSD symptoms for clinical diagnoses. This meta-analysis study demonstrates the potential for affecting PTSD symptoms using EEG. Further research to establish a gold standard protocol for EEG-based neuro-feedback training for PTSD symptoms is needed.

## Figures and Tables

**Figure 1 ijerph-19-13096-f001:**
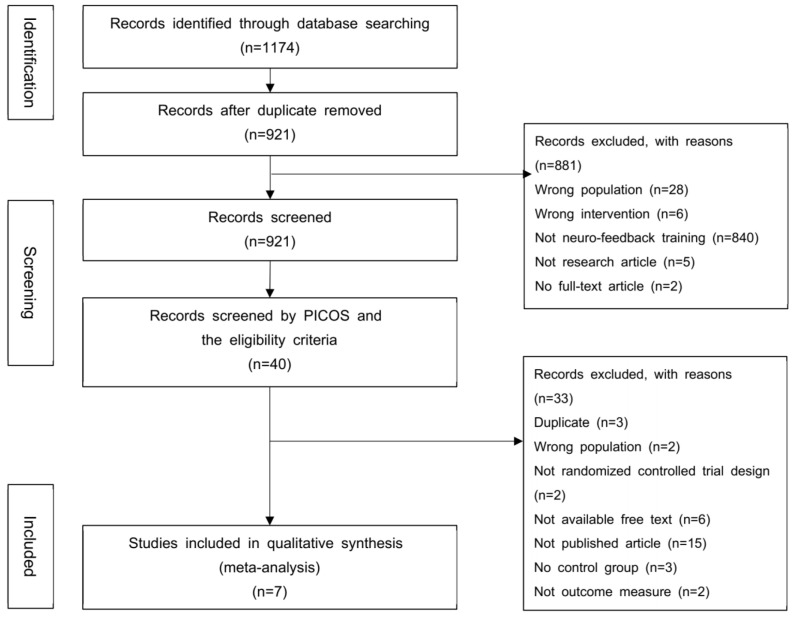
Flowchart of studies included in the meta-analysis.

**Figure 2 ijerph-19-13096-f002:**
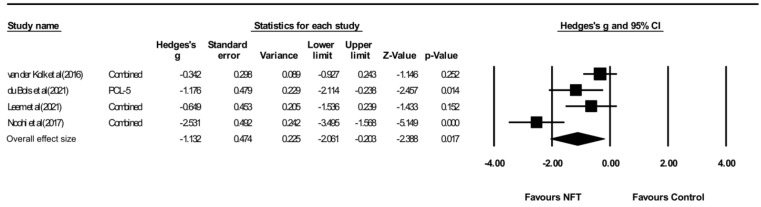
Forest plot of meta-analysis of EEG-NFT on overall PTSD symptoms. vander Kolket et al (2016) [29] du Bois et al (2021) [30]; Leem et al (2021) [31]; Noohi et al (2017) [32].

**Figure 3 ijerph-19-13096-f003:**
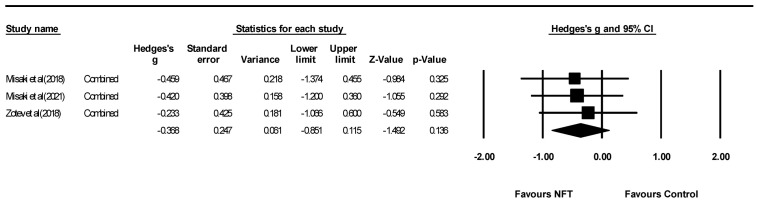
Forest plot of meta-analysis of fMRI-NFT on overall PTSD symptoms. Misaki et al (2018) [26]; Misaki et al (2021) [27]; Zotev et al (2018) [28].

**Figure 4 ijerph-19-13096-f004:**
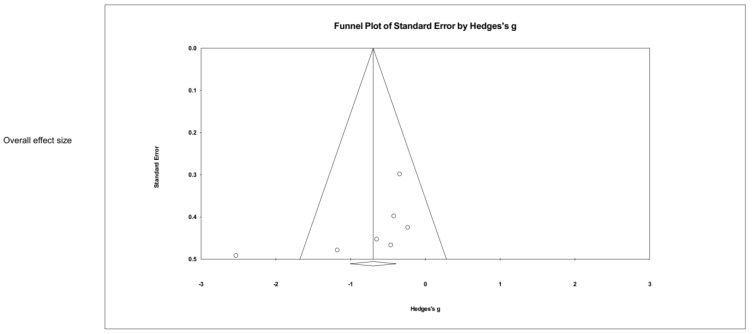
Funnel plot for meta-analysis of the effect of neuro-feedback training on overall PTSD.

**Figure 5 ijerph-19-13096-f005:**
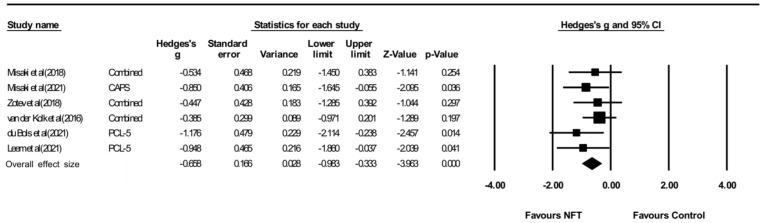
Forest plot for meta-analysis of the effect of neuro-feedback training on PTSD diagnosis. Misaki et al (2018) [26]; Misaki et al (2021) [27]; Zotev et al (2018) [28]; vander Kolket et al (2016) [29] du Bois et al (2021) [30]; Leem et al (2021) [31].

**Figure 6 ijerph-19-13096-f006:**
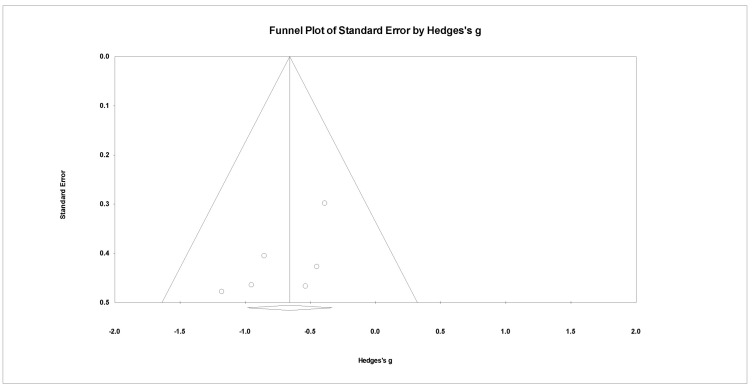
Funnel plot for meta-analysis of the effect of neuro-feedback training on PTSD diagnosis.

**Figure 7 ijerph-19-13096-f007:**
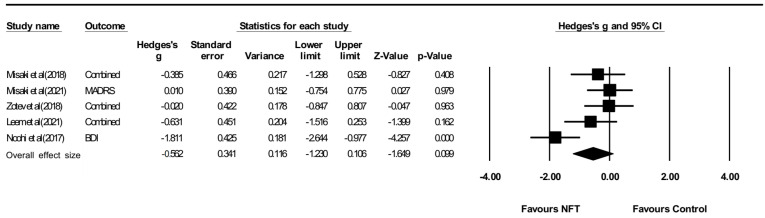
Forest plot for meta-analysis of the effect of neuro-feedback training on anxiety and depression. Misaki et al (2018) [26]; Misaki et al (2021) [27]; Zotev et al (2018) [28]; Leem et al (2021) [31]; Noohi et al (2017) [32].

**Figure 8 ijerph-19-13096-f008:**
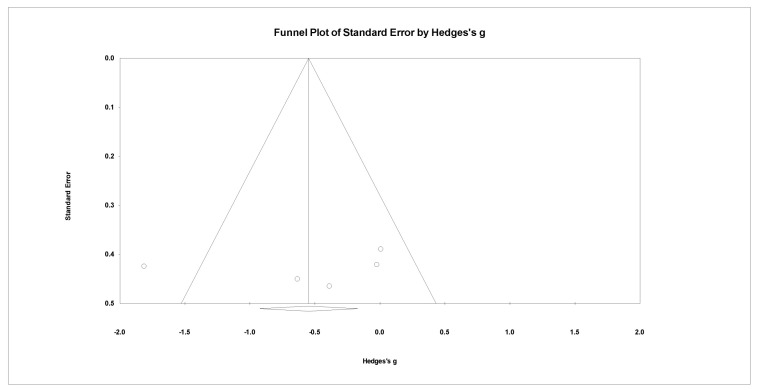
Funnel plot for meta-analysis of the effect of neuro-feedback training on anxiety and depression.

**Figure 9 ijerph-19-13096-f009:**
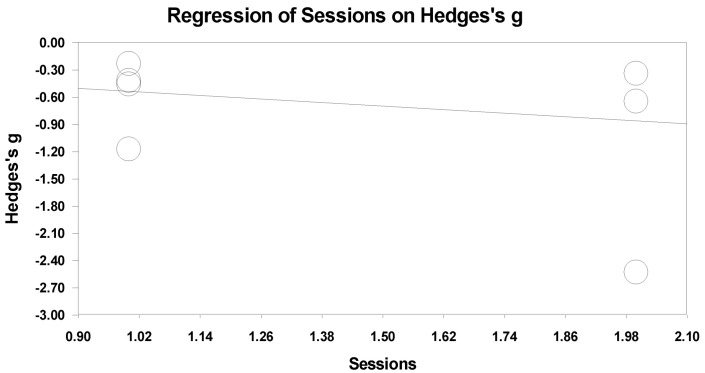
Meta-regression analysis of training sessions on Hedges’ s g.

**Figure 10 ijerph-19-13096-f010:**
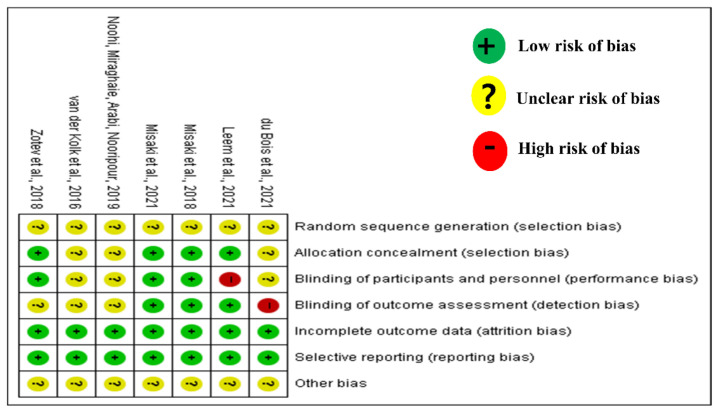
Summary of Cochran risk of bias in RCTs. Green indicates low risk of bias, yellow indicates unclear bias, and red indicates high bias risk. Misaki et al (2018) [26]; Misaki et al (2021) [27]; Zotev et al (2018) [28]; vander Kolket et al (2016) [29] du Bois et al (2021) [30]; Leem et al (2021) [31].

**Table 1 ijerph-19-13096-t001:** Characteristics of studies included in Meta-analysis.

Reference	Study Design	Sample				NFT Characteristics			Outcomes
		Experimental Group		Control Group		Type	Modality	Sessions	
		N	Age (m ± SD)	N	Age (m ± SD)				
Misaki et al., 2018 [26]	RCT	16	30 ± 6	6	31 ± 9	fMRI	Left amygdala	3 sessions (6 min 50 s); visit 6, visit 7 follow-up	CAPS, PCL-M, MADRS, HAM-A
Misaki et al., 2021 [27]	RCT	20	21–48	9	21–48	fMRI	Hippocampal volume in the left amygdala	3 sessions (6 min 50 s); visit 6, visit 7 follow-up	CAPS, MADRS
Zotev et al., 2018 [28]	RCT	15	31 ± 5	8	37 ± 8	fMRI with simultaneous EEG procedure	Left amygdala	3 sessions (8 min 46 s); visit 8	CAPS, PCL-M, HDRS, MADRS
Van der Kolk et al., 2016 [29]	RCT	28	46.04 ± 12.89	24	42.45 ± 13.5	EEG	T4, P4, A1; alpha	24 sessions (30 min); 12 weeks, 16 week one month follow-up	CAPS, DTS, IASC
Du Bois et al., 2021 [30]	RCT	10	-	9	-	EEG	Pz channel: alpha (8–12 Hz)	7 sessions (134 s)	PCL-5
Leem et al., 2021 [31]	RCT	10	44.40 ± 13.61	9	43.56 ± 19.1	EEG	Pz channel: alpha (8–12 Hz)-theta(4–7 Hz)	16 sessions (10 min); 8 weeks, 12 week one month follow-up	PCL-5, IES-R, BAI, BDI
Noohi et al., 2017 [32]	RCT	15	25–60	15	25–60	EEG	Mid and Frontal areas; alpha-theta ratio	25 sessions (30–40 min); four time week; after 45 days follow-up	IES-R, BDI

Note: RCT = randomized controlled trial; fMRI = functional magnetic resonance imaging; EEG = electroencephalogram; CAPS = Clinician-Administered PTSD Scale; PCL-M = PTSD Checklist-Military Version; MADRS = Montgemery–Asberg Depression Rating Scale; HAM-A = Hamilton Anxiety Scale; PCL-5 = PTSD Diagnostic and Statistical Manual of Mental Disorders Checklist-5th edition; HDRS = Hamilton Depression Rating Scale; DTS = the Davidson Trauma Scale; IASC = Inventory of Altered Self-Capacities; IES-R = Impact of the Event Scale-revised; BAI = Beck Anxiety Inventory; and BDI = Beck Depression Inventory.

## Data Availability

Not applicable.

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
