# Peer review of "Efficacy of Neuro-Feedback Training for PTSD Symptoms: A Systematic Review and Meta-Analysis"

_ijerph, 2022, doi:10.3390/ijerph192013096_

Round 1

Reviewer 1 Report

Dear Authors,

You will find my comments and suggestions in the file attached.

Author Response

Thank you for your constructive comments. We have tried our best to reflect your comments on our revised manuscript. We have attached our reply to your comments.

Reviewer 2 Report

The review manuscript ‘Efficacy of Neuro-feedback Training for PTSD Symptoms: A Systematic Review and Meta-Analysis’ utilizes a meta-analysis to study the neuro-feedback training for PTSD symptoms and PTSD severity assessment, showing that EEG-NFT was more helpful for training PTSD symptoms than fMRI-NFT and PTSD severity was ameliorated but not psychological symptoms such as anxiety and depression after NFT. The review and analysis part were well performed except the following comments that needs to be improved by the authors:

1.     Line 141, authors should claim each number of records from different database research according to their listing in Method part: PubMed, the Cochrane Library, Web of Science, Science Direct, and ClinicalTrials.gov databases.

2.     Figure5, 6 and 7 were not correctly cited. Authors should double check the citation of the manuscript before submission.

3.     In the discussion part, authors should list in detail about the limitation of the meta-analysis but not only discuss the statistical findings on the involved research.

4.     Authors only involved EEG-NFT or fMRI-NFT on PTSD symptoms and prognosis, is there any research regarding combining EEG and fMRI-NFT and what’s the difference?

5.     Line 186, PTSD severity should be introduced before but not only listed in the result part.

Author Response

(The authors gave the same response as above.)

Reviewer 3 Report

The authors of this study aimed to find the impact of neuro-feedback training (NFT) on post-traumatic stress disorder PTSD symptoms. Their study focused on the efficacy of functional magnetic resonance imaging (fMRI)-based NFT (fMRI-NFT) and electroencephalogram (EEG)-based NFT (EEG-NFT) on PTSD symptoms. They conducted a systematic meta-analysis based on seven studies that met the inclusion criteria. Their findings showed that EEG-NFT was more helpful for training PTSD symptoms than fMRI-NFT.

Specific comments:

The authors performed this study that answered the research question.

1.     Overall, this paper needs minor editing of grammar and language before being published.

2.     I think the introduction is short and does not provide sufficient background information.

3.     It appears that the figure 1 presentation is not very clear. It is very hard to read the text within the flow chart.

4.     Importantly, it is unclear whether all of these seven studies have excluded biases in determining psychological symptoms, including anxiety and depression. Also, it is unclear what standards they followed to measure anxiety and depression as these symptoms are very subjective.

Author Response

(The authors gave the same response as above.)

Round 2

Reviewer 1 Report

Dear authors, 

You will find our comments and suggestions in the file attached.

Author Response

Thank you for your sharp observation. We apologize for the incorrect values in the text. We have revised them and added the limitation due to the small number of the included studies in this meta-analysis.
